# How to speed up ion transport in nanopores

Konrad Breitsprecher[1], Mathijs Janssen [2,3,8], Pattarachai Srimuk[4,5], B. Layla Mehdi [6], Volker Presser [4,5✉], Christian Holm [1] & Svyatoslav Kondrat [2,3,7✉]

Electrolyte-filled subnanometre pores exhibit exciting physics and play an increasingly important role in science and technology. In supercapacitors, for instance, ultranarrow pores provide excellent capacitive characteristics. However, ions experience difficulties in entering and leaving such pores, which slows down charging and discharging processes. In an earlier work we showed for a simple model that a slow voltage sweep charges ultranarrow pores quicker than an abrupt voltage step. A slowly applied voltage avoids ionic clogging and co-ion trapping—a problem known to occur when the applied potential is varied too quickly— causing sluggish dynamics. Herein, we verify this finding experimentally. Guided by theoretical considerations, we also develop a *non-linear* voltage sweep and demonstrate, with molecular dynamics simulations, that it can charge a nanopore even faster than the corresponding optimized linear sweep. For discharging we find, with simulations and in experiments, that if we reverse the applied potential and then sweep it to zero, the pores lose their charge much quicker than they do for a short-circuited discharge over their internal resistance. Our findings open up opportunities to greatly accelerate charging and discharging of subnanometre pores without compromising the capacitive characteristics, improving their importance for energy storage, capacitive deionization, and electrochemical heat harvesting.

[1] Institute for Computational Physics, Universität Stuttgart, Allmandring 3, 70569 Stuttgart, Germany. [2] Max-Planck-Institut für Intelligente Systeme, Heisenbergstrasse 3, 70569 Stuttgart, Germany. [3] IV. Institut für Theoretische Physik, Universität Stuttgart, Pfaffenwaldring 57, 70569 Stuttgart, Germany. [4] INM - Leibniz Institute for New Materials, Campus D2 2, 66123 Saarbrücken, Germany. [5] Department of Materials Science and Engineering, Saarland University, Campus D2 2, 66123 Saarbrücken, Germany. [6] University of Liverpool, School of Engineering, 514 Brodie Hall, L69 3GQ Liverpool, UK. [7] Department of Complex Systems, Institute of Physical Chemistry, PAS, Kasprzaka 44/52, 01-224 Warsaw, Poland. [8] Present address: Mechanics Division, Department of Mathematics, University of Oslo, 0316 Oslo, Norway. ✉email: volker.presser@leibniz-inm.de; svyatoslav.kondrat@gmail.com

Electrolyte-immersed porous electrodes are an essential building block for many state-of-the-art technologies in energy storage[1–4], energy harvesting[5–7] and capacitive deionization[8–10]. Of particular importance are electrical double-layer capacitors (EDLCs), often called supercapacitors or ultra-capacitors[1–4,6,11,12], which store energy by electrosorption of ionic charge into porous electrodes. Their performance is characterized by the energy density $\mathcal{E}$ and power density $\mathcal{P}$, which describe the amount of energy and the speed with which it can be supplied to an external load or device. Owing to their particular $(\mathcal{E}, \mathcal{P})$ properties, supercapacitors find their way in applications that need higher power than delivered by batteries and more energy than stored in traditional dielectric capacitors. So far, the highest achievable capacitance[13–15] and energy densities[16] have been obtained with nanometre-sized pores. Hence, an extensive research effort has been dedicated to understanding the properties of such nanoporous supercapacitors[12,17–27], with an overarching goal being to maximize the stored energy density[28–35] and the speed of charging and discharging[36–44].

It is often assumed that charging and discharging times are equal and proportional to the time constant $\mathcal{E}/\mathcal{P}$[2,45]. This reasoning probably originates from simple electronic $RC$ circuits, which indeed display a charge/discharge symmetry (Supplementary Note S1). However, even for a simple EDLC model with planar electrodes, such charge/discharge symmetry is not present for applied potentials above the thermal voltage ($\approx 25$ mV at room temperature)[46–48]. Molecular dynamics (MD) simulations have demonstrated that also nanopores charge and discharge in a dissimilar manner[41]. The knowledge of discharging behaviour is thus insufficient for predicting charging behaviour and vice versa. Therefore, in this article, which centres around charge-discharge time optimization, we consider charging and discharging separately. Furthermore, as charge and discharge times are affected by many parameters, any optimization study should specify which parameters are varied and which are kept fixed. One could ask, for instance, given a charging or discharging procedure (voltage step or sweep, etc.), which supercapacitor charges/discharges the fastest? This is a problem of minimizing internal resistances via different pore shapes, lengths, etc.[41,49]. An alternative question that one could ask is: given a supercapacitor, which time-dependent cell voltage minimizes the time spent to charge it to a certain charge $Q$ and to discharge the same supercapacitor to $Q = 0$? In this study, we focus on the second question, which has received much less attention in electrochemistry[42]. In other contexts, optimization of time-dependent protocols driving a system from an initial to a final state has been investigated, for instance, to find shortcuts to adiabaticity in quantum systems[50,51], to engineer swift equilibration of Brownian particles[52] and AFM tips[53], etc. Finding the voltage sweeps providing the shortest charging times is of obvious practical importance to supercapacitors. Finding optimal discharging sweeps, which we also set out to do, is relevant to electrochemical low-grade heat harvesting[7] and capacitive deionization[8–10], where operation speed may be more important than energy efficiency.

Recent MD simulations have shown that, when a potential difference is applied abruptly to a nanopore, counterions rush to the pore's entrance and clog the pore, causing co-ion trapping deep in its interior and leading to an overall sluggish charging dynamics[41,54,55]. Pore clogging can be avoided by applying the potential difference with a linear sweep, i.e., by varying it with a constant rate[41] (Fig. 1e). We show in this study that charging can be made even faster than the fastest (optimized) linear sweep developed in ref. [41], if the variation of the applied potential is matched to the actual rate of co-ion desorption. We propose a general expression for a non-linear sweep function and consider two closed-form approximations. We assess the benefits of such a non-linear voltage sweep in MD simulations of a model supercapacitor and in experiments with novolac-derived carbon electrodes featuring a narrow pore-size distribution and mainly slit-shaped pores (Fig. 1b and Supplementary Fig. S3).

Finally, for discharging, ref. [41] found that a step voltage unloads a supercapacitor faster than any finite-rate linear potential sweep. Here we demonstrate, with MD simulations and experiments, that a supercapacitor can discharge even faster if we apply a non-linear sweep consisting of a voltage inversion followed by a linear sweep to zero.

## Results

**Non-linear voltage sweep to accelerate charging.** To avert the catastrophic clogging of a nanopore subject to a large instantaneous voltage step, the time-dependent potential difference should be increased gently, so that the co-ions can leave the pore before counterions clog it. For a tiny increment $\Delta U$ of the applied potential—much too small to yield pore clogging—the co-ion desorption time can be estimated as $\Delta t = l^2/D_c$, where $D_c$ is the in-pore co-ion diffusion constant and $l$ the distance of the co-ion to the pore exit. Then the rate of an optimal *linear* sweep can be approximated by $k_{opt} = \Delta U/\Delta t_{max}$, with $\Delta t_{max}$ being the longest of these times[41]. However, as $\Delta t$ may differ from $\Delta t_{max}$ at different stages of charging, it may be possible to charge a nanopore faster than with the optimized linear sweep, by varying $\Delta t$, and hence the sweeping rate $k$, in the course of charging.

*Derivation of the non-linear sweep function.* Before a potential difference is applied to a nanopore, the distribution of co-ions in a pore is homogeneous. We can split these co-ions into $K_0$ ($\approx \rho_c^0 w \ell a$) imaginary 'layers' along the pore, where $\rho_c^0$ is the in-pore co-ion density at zero voltage, $w$ is the pore width, $\ell$ is the pore length and $a$ is the ion diameter, for simplicity assumed the same for cations and anions. We then apply the potential difference stepwise such that in equilibrium each voltage step would lead to exactly one fewer layer of co-ions in the nanopore. After each step, we keep the voltage constant until this one layer diffuses out. It is noteworthy that the system does not have to equilibrate fully during this time. For instance, at low voltages it might be sufficient that only the co-ions close to the pore exit redistribute in response to a voltage step. Provided the co-ion density is roughly homogeneous at the end of each waiting period, we can estimate the average distance over which the co-ions have to diffuse out after the $n$th voltage step as $l = \ell/(K_0 - n + 1)$. The waiting time $\Delta t_n$ after the $n$th step can now be estimated as the time needed by a co-ion layer to diffuse out of the pore:

$$\Delta t_n = \frac{1}{D_c(n)} \frac{\ell^2}{(K_0 - n + 1)^2}, \quad (1)$$

where the co-ion diffusion coefficient $D_c$ may depend on $n$. The total time $t$ needed to charge a nanopore to a potential difference $U$ is the sum of all time increments up to $n = K_0 - K(U)$, where $K(U)$ is the equilibrium number of layers in the pore at $U$, i.e., $t(U) = \sum_{n=0}^{n < K_0 - K(U)} \Delta t_n$. For example, taking a constant $D_c = 10^{-8}$ m$^2$ s$^{-1}$ and $\ell = 12$ nm ($K_0 \approx 8$, cf. Fig. 2), we find $t(U) \approx 8$ ns for the voltage $U$ at which the pore becomes free of co-ions. In this sum, the waiting time $\Delta t_n$ increased from $\Delta t_0 \approx 0.18$ ns after the first step to $\Delta t_7 \approx 3.6$ ns after the last step, a more than ten-fold increase due to the increased distance $l$ that the co-ions needed to cover to exit a pore with fewer and fewer co-ions.

Using the above voltage steps that each expels one layer of co-ions gives a crude procedure to charge a pore as fast as possible. We can refine this procedure by using many tiny voltage steps, each expelling some small amount of co-ions instead. Accordingly, we rewrite the sum over $n$ as an integral and change the

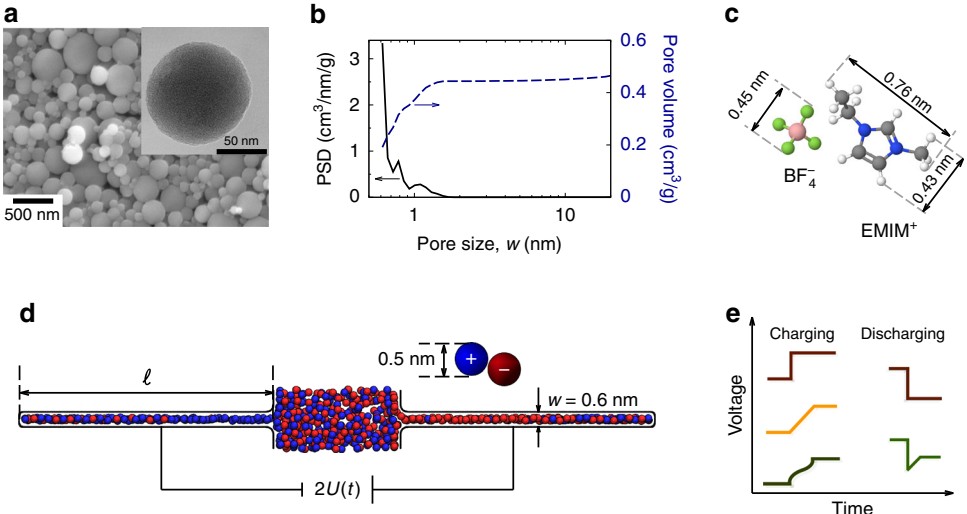

**Fig. 1 Systems and charge-discharge voltage sweeps. a** SEM and BF-STEM (the inset) images of novolac-derived porous carbon particles. **b** Pore-size distribution (PSD) and cumulative PSD of the novolac-derived porous electrodes. **c** Schematic drawings and sizes of EMIM-BF$_4$ ionic liquid used in the experiments. **d** Simulation model of a supercapacitor and an ionic liquid. Ions were modelled as charged spheres and each electrode consisted of one slit nanopore. The accessible pore width $w = 0.6$ nm and the ion diameter $a = 0.5$ nm have been used in all simulations. The pore length $\ell$ was 12, 16, and 20 nm. **e** Schematics of cell voltages used in this study for charging and discharging supercapacitors.

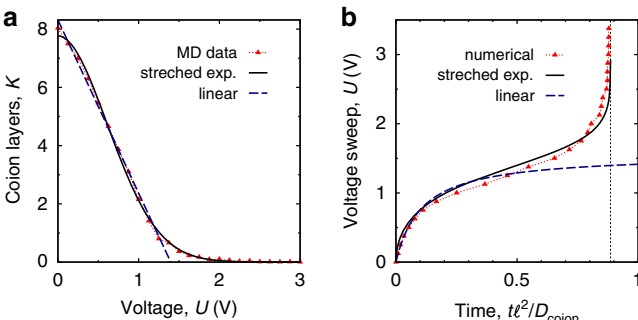

**Fig. 2 Determination of non-linear sweep function. a** Number of co-ion layers in a nanopore, $K$, as a function of potential $U$ applied to the pore with respect to the bulk electrolyte. Symbols denote the values obtained from MD simulations of a model supercapacitor (Fig. 1d). $K$ was calculated as $K = \rho_c w \ell a = N_c a/h$, where $\rho_c = N_c/(w \ell h)$ and $N_c$ are the density and the number of co-ions in a pore, respectively, $a$ is the ion diameter, and $w$, $\ell$, and $h$ are the width, length, and height of the pore; $K_0$ is the value of $K$ at $U = 0$ V. The solid and dashed lines are approximations to these data by stretched exponential (eq. (3)) and linear (eq. (7)) functions, respectively. The ion diameter $a = 0.5$ nm, the pore width $w = 0.6$ nm and the pore length $\ell = 12$ nm; for other pore lengths see Supplementary Figs. S5 and S6, and for the fitting parameters see Fig. S7. **b** Non-linear voltage sweeps as obtained by numerically integrating eq. (2) with the MD data for $K(U)$ from **a** and by using the stretch exponential and linear approximations for $K(U)$; the corresponding non-linear sweep functions are given by eqs. (6) and (9), respectively. The thin dotted line shows $t = \tau$, where $U(t)$ diverges, see eqs. (5) and (6).

integration variable from $n$ to voltage $u$, which yields:

$$t(U) \approx -\ell^2 \int_0^U \frac{(dK/du)}{D_c(u)[K(u)+1]^2} \, du. \tag{2}$$

Equation (2) is an implicit equation for the time-dependent sweep function $U(t)$ in terms of $K(U)$. This equation provides the optimal sweep function for charging to an arbitrary potential difference below $U$. To demonstrate its applicability, we have numerically determined $K(U)$ for the model supercapacitor

shown in Fig. 1d with MD simulations; the results are shown by symbols in Fig. 2a. Assuming that $D_c$ does not depend on the applied potential, we performed the integration in eq. (2) numerically using the composite trapezoidal rule (symbols in Fig. 2b; see also Figs. S5 and S6). The results demonstrate that $U(t)$ can be varied quickly at early times, corresponding to low voltages, because there are still many co-ions close to the pore exit, and hence their 'effusion' path is relatively short (i.e., the path over which the co-ions need to diffuse to exit the pore). At intermediate times, corresponding to intermediate voltages, this effusion path becomes longer due to the reduced co-ion density and one must slow down the variation of the applied potential, to allow the co-ions to leave the pore. Finally, at long times (higher applied potentials), there are practically no co-ions left in the pore and the charging can proceed in an almost stepwise manner.

*Stretched exponential approximation.* To complement the above insights from the numerical evaluation of eq. (2), we approximated the simulation data of $K(U)$ by a stretched exponential:

$$K(U) = K_0 \exp[-(\gamma U)^\alpha], \tag{3}$$

allowing us to perform the integral in eq. (2) analytically. Here, $K_0$, $\gamma$, and $\alpha$ are $\ell$-dependent fitting parameters. We have used the dogleg least-square algorithm from scipy (https://scipy.org), to fit eq. (3) to the MD data. Figure 2a demonstrates that eq. (3) fits the MD data decently (solid line in Fig. 2a, see also Figs. S5, S6, and S7). Notably, we found that the fitting parameters $\alpha$ and $\gamma$ depend weakly on the pore length, $\ell$, whereas $K_0$ varies with $\ell$ more significantly (viz. , $K_0 \sim \ell$, see Fig. S7). This is understandable as $\alpha$ and $\gamma$ define the shape of $K(U)$, which does not change appreciably with $\ell$ (Fig. S7), whereas $K_0$ represents the number of co-ion layers in the pore at $U = 0$ V, which should indeed increase with the pore length.

Plugging eq. (3) into eq. (2) and assuming again that $D_c$ is $U$ independent, one finds:

$$t(U) = \tau \frac{1 - K(U)/K_0}{1 + K(U)}, \tag{4}$$

where:

$$\tau = \frac{\ell^2}{D_c} \frac{K_0}{K_0 + 1}. \tag{5}$$

Inverting eq. (4) leads to:

$$U(t) = \gamma^{-1} \left( \ln \frac{1 + K_0 t/\tau}{1 - t/\tau} \right)^{1/\alpha}. \tag{6}$$

Figure 2b shows that eq. (6) provides a good approximation to the non-linear sweep function obtained by integrating eq. (2) numerically. For very long pores, $\ell \gg (\rho_c^0 wa)^{-1}$, one has $K_0 \sim \ell \gg 1$ and eq. (5) gives $\tau \sim \ell^2$ in the leading order in $\ell$. Then eq. (4) leads to charging times $t(U) \sim \ell$ at potential differences below threshold $U_t = \gamma^{-1}(\ln K_0)^{1/\alpha}$ (at which the number of co-ion layers $K(U) \approx 1$) and $t(U) \sim \ell^2$ at $U \gg U_t$. The latter result is in line with the quadratic pore-length scaling of the charging times of optimal linear sweep[41]. For the pore-lengths considered in the simulations ($\ell = 12$, 16, and 20 nm), the threshold voltage $U_t$ varies slightly around 1.5 V and increases roughly as a square root of logarithm with increasing $\ell$.

*Linear approximation.* Although the stretched exponential (eq. (3)) approximates the number of co-ion layers $K(U)$ for all applied potentials considered (Fig. 2a), the resulting expression for the non-linear sweep function, eq. (6), looks cumbersome. A useful approximation, likely most relevant to experimental systems[24,26,56], can be obtained for low voltages, $U \lesssim 1.3$ V in Fig. 2a. We notice that $K(U)$ varies roughly linearly in this regime, i.e.,

$$K(U) = K_0(1 - \gamma U), \tag{7}$$

where $\gamma$ and $K_0$ are again $\ell$-dependent fitting parameters. We have fitted eq. (7) to the MD data for $U \lesssim 1.3$ V (dashed line in Fig. 2a and Fig. S8) using the dogleg least-square algorithm from scipy (https://scipy.org).

Plugging eq. (7) into eq. (2) and assuming $D_c$ to be $U$ independent, as before, one obtains:

$$t(U) = \frac{\tau \gamma U}{1 + K(U)}, \tag{8}$$

where $\tau$ is given by eq. (5). Inverting eq. (8) yields a simple equation:

$$U(t) = \frac{U_0 \, t}{t + \tilde{\tau}} \tag{9}$$

where $U_0 = (1 + K_0)/(\gamma K_0)$ and $\tilde{\tau} = \tau/K_0$. It is interesting to note that the linear approximation for $K(U)$, eq. (7), still yields a sweep function, eq. (9), which varies non-linearly with time.

Figure 2b shows that eq. (9) approximates the non-linear sweep function (obtained by numerical integration of eq. (2)) well at low voltages and poorly at high voltages. This is not surprising, as we fitted the linear approximation only in the low-voltage regime.

For long pores, $\ell \gg 1/(\rho_c wa)$, one has $\tau \sim \ell^2$, and eq. (8) gives $t(U) \approx [\ell^2/(D_c K_0)][\gamma U/(1 - \gamma U)]$. Since $K_0 \sim \ell$ and $\gamma$ depends only weakly on $\ell$ (Fig. S8), the charging time increases proportionally to the pore length, $\ell$. This linear $\ell$-scaling is in line with the scaling found for the stretched exponential approximation at low applied potentials. Conversely, for the optimal linear sweep at high voltages the charging time scales as $\ell^2$[41].

*Results of MD simulations.* To assess the benefits of the non-linear sweeps, as compared to step voltage and optimal linear sweeps[41], we performed MD simulations, in which we applied these three different charging protocols to the model supercapacitor shown in Fig. 1d. In all simulations, we chose a large potential difference

$U = 3$ V. The reason for this choice is that, at this $U$, co-ion trapping, which we aim to circumvent with slow voltage sweeps, occurs for relatively short, computationally feasible pores. For longer pores the trapping and pore clogging occur at lower potential differences (Fig. S4), suggesting that our approach applies to a wider voltage range.

For the non-linear sweep function, we used eq. (6), as obtained by fitting the stretched exponential approximation to the MD data for $K(U)$, the number of co-ion layers at equilibrium (Fig. 2). Equation (6) depends parametrically on $\tau$, which, in turn, depends on the unknown co-ion diffusion coefficient $D_c$ (eq. (5)). Accordingly, we treated $\tau$ as an optimization parameter and performed MD simulations of the out-of-equilibrium charging behaviour with the non-linear sweep functions for several values of $\tau$. Examples of $U(t)$ for $\tau = 4$ ns and $\tau = 12$ ns are shown in Fig. 3a. Figure 3a also shows the accumulated charge and the number of counter and co-ions inside the pore as a function of time. Clearly, the non-linear sweep provides faster charging than the step voltage for all $\tau$ considered. Only for some $\tau$, the non-linear sweep charges faster than the optimal linear sweep. When sweeps are too fast (small $\tau$), co-ion trapping leads to sluggish desorption, as is the case for the step-voltage charging or too-fast linear sweeps (red and dark-green curves in Fig. 3a). For sufficiently large $\tau$, the co-ion desorption proceeds faster than for the optimal linear sweep, as anticipated (orange vs. light-green curves in Fig. 3a).

To determine the optimal $\tau$, we studied the $\tau$ dependence of the co-ion desorption and counterion adsorption times, which are the times needed by the co-ion and counterion densities to reach their equilibrium values corresponding to the final applied voltage (these times are not the same[41] and depend on charging protocol $U(t)$; see Figs. S10 and S11). The desorption time increases drastically for small $\tau$, which is due to co-ion trapping, and levels off for larger values of $\tau$, as Fig. 3b demonstrates. The adsorption time increases roughly linearly with $\tau$ for $\tau > 8$ ns, but increases abruptly for smaller values of $\tau$ (the dashed line in Fig. 3b). Figure 3b also shows that the optimal $\tau$-value, $\tau_{opt}$, i.e., the value that minimizes the charging time, can be found as a crossing point of adsorption and desorption times (there might be more than one crossing; clearly, $\tau_{opt}$ corresponds to the one providing the shortest time). For sufficiently slow charging, $\tau \geq \tau_{opt}$, the charging times from the MD simulations are consistent with the charging times given by eq. (4), i.e., the pore charging completes at the same time as the voltage reaches its final value ($\tau = 12$ ns in Fig. 3a and Fig. S11). For lower values of $\tau$, the voltage varies too quickly, which leads to pore clogging and causes slow co-ion desorption ($\tau = 4$ ns in Fig. 3a and Fig. S10).

Next, we performed MD simulations for a few pore lengths $\ell$ and determined $\tau_{opt}$ in each case. From the accumulated charge $Q$ (obtained at $\tau_{opt}$), we extracted the charging time, $t_{charge}$, which we defined as the time at which $Q$ reaches 96% of the maximum charge capacity at a given applied potential (note that the so determined $t_{charge}$ is only approximately equal to $t(U)$, eq. (4), as we allowed for 4% tolerance in co-ion desorption, to be consistent with ref. [41]). The results are shown in Fig. 3c, along with the charging times obtained with the optimal linear sweep[41]. The difference between the charging times for the linear and non-linear sweep functions increases markedly with the pore length, $\ell$. The data suggest that the non-linear sweeps may provide significantly faster charging for longer pores.

*Experimental results.* To test our theoretical findings, we have performed charging experiments with a supercapacitor based on a symmetric two-electrode setup and novolac-derived activated carbons with well-controlled nanopores as the electrode material[57] (Fig. 1a, b). As seen from the gas sorption data, the pore size

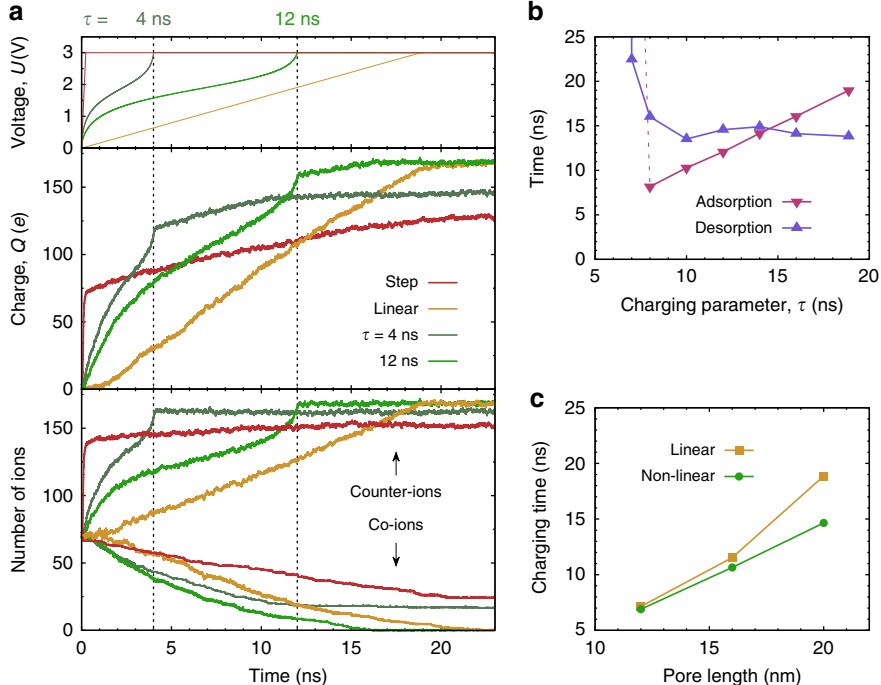

**Fig. 3 Non-linear sweeps versus step-voltage and optimal linear sweep. a** Potential $U(t)$ applied to a nanopore with respect to the bulk electrolyte (top), accumulated charge (middle) and the number of ions (bottom) for step-voltage, optimized linear and two non-linear sweep functions from MD simulations. $U(t)$ for non-linear sweeps have been obtained via fitting the MD data to the stretched exponential (see eq. (3) and Fig. 2), and $\tau$ is an 'optimization' parameter, see eqs. (5) and (6). For comparison of the non-linear and linear sweeps with the same stopping time $t = 12$ ns, see Fig. S9. **b** Counterion adsorption and co-ion desorption times as functions of parameter $\tau$ (see Figs. S10 and S11). The optimal $\tau$ value, $\tau_{opt}$, corresponds to $\tau$ at which the adsorption and desorption times intersect. **c** Charging times for linear and non-linear sweeps as functions of pore length. In **a** and **b**, the pore length $\ell = 20$ nm. In all plots, the ion diameter $a = 0.5$ nm and the pore width $w = 0.6$ nm.

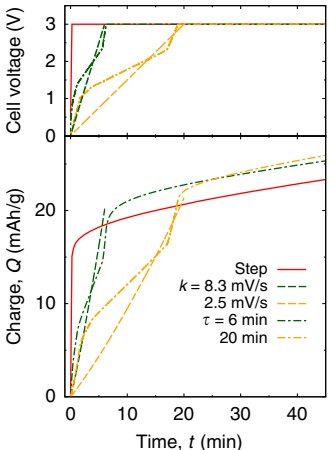

**Fig. 4 Non-linear sweeps versus step-voltage and linear sweeps for novolac-derived carbons.** Cell voltage (top) and accumulated charge (bottom) for step-voltage, linear, and non-linear sweep charging. For the non-linear sweeps, cell voltage vs time have been obtained by using the stretched exponential approximation (see eq. (3)), for two values of $\tau$, as indicated on the plot. The linear sweeps have been obtained via a one-cycle voltammetry; the discharging parts of these curves are not shown for clarity.

distribution is very narrow with an average pore size of 0.68 nm and a total pore volume of 0.49 cm$^3$/g. We used EMIM-BF$_4$ ionic liquid as electrolyte (Fig. 1c). All experiments were conducted at room temperature (including discharging, see below). However, to check how our results depend on the IL conductivity, we carried out a series of additional experiments at an elevated

temperature, at which the bulk conductivity was doubled (Fig. S12). At both temperatures we observed qualitatively the same results (Fig. 4 and Fig. S13).

We first applied a step-like cell voltage to the supercapacitor and measured how the accumulated charge varies over time (the solid red lines in Fig. 4). We found that the charge can be fitted decently by the sum of two exponents[38], $Q(t) = Q_\infty[1 - a_1 \exp(-t/\tau_1) - a_2 \exp(-t/\tau_2)]$ (for the fitting parameters and the plot see Fig. S14). The slow timescale $\tau_2 \approx 45$ min is in agreement with model predictions saying that supercapacitors charge at late times with the timescale $\tau = \alpha(L/2 + H)^2/D \approx 17$ min[58], where $L \approx 150$ μm is the electrode separation (which in the experiments is the thickness of the electrode–electrode separator), $H \approx 109$ μm is the electrode thickness, $D \approx 10^{-11}$ m$^2$/s the bulk diffusion constant of EMIM-BF$_4$. The salt concentration prefactor $\alpha = 0.59$ was determined for the case of a model RTIL with ionic radii the same as in our simulations (0.5 nm) (private communication with Cheng Lian). As our supercapacitor model contains just two pores and a nanometre-sized reservoir (Fig. 1d), and hence does not account for the multiscale nature of real supercapacitors, this slow diffusive response is absent in our MD simulations. Conversely, the model of ref. [58] does not describe the dynamics of finite size ions in nanopores in as much detail as our simulations do, and hence, ref. [58] could not account for the pore clogging effects central to our article.

We next studied how this abrupt step-voltage charging compares with slower voltage sweeps. We generated non-linear sweep data in a computer for a few values of $\tau$ according to eq. (6), similarly as in the simulations. The non-linear sweeps were produced by a piecewise approximation via linear functions (dash-dot lines in the top plot of Fig. 4). To study the linear-sweep charging, we used a 'single-cycle' voltammetry; the single

run, rather than cycling, was necessary to ensure that the supercapacitor was fully discharged initially. The sweep rates $k$ were chosen such that the times $\tau_k = U/k$ ($U = 3$ V is the applied cell voltage) were comparable to the values of $\tau$ used in the non-linear sweeps (in the voltammetric experiments, the cell voltage was swept to zero and the device was allowed to discharge after reaching the final value of 3 V. For clarity, we show only the charging parts of these $Q(t)$ curves in Fig. 4.)

Figure 4 shows that the voltage step is overtaken by all four slower applied potentials. This constitutes the first experimental verification of ref. [41], which suggested that a step-voltage does worse than linear sweep charging. We note, however, that both the linear and non-linear sweeps have not been optimized and hence, unlike in the simulations (Fig. 3), the charge in Fig. 4 continues to grow after the cell voltage reaches its final value.

Although our experiments have demonstrated that a slow variation of applied potential can make charging faster than the step-voltage, they were inconclusive regarding the comparison of the linear and non-linear sweeps (we note, however, that the non-linear sweep with $\tau = 20$ min performed slightly better than the corresponding linear sweep with $k = 2.5$ mV/s, which is consistent with the simulations). This is likely because we have not reached the optimal sweep rate, which appeared to be slower than 0.1 mV/s (Fig. S15), implying that the time needed to fully charge the supercapacitor was >8 h. Thus, the voltage sweeps in the experiments operated only in short times (compare with $\tau = 4$ ns in Fig. 3a). Nevertheless, both the simulations and the experiments clearly demonstrate that 'slow' voltage sweeps have the potential to speed up charging.

**Accelerating discharging by voltage inversion**. Turning to the discharging of supercapacitors, we are interested in the quickest discharging voltage sweep $U(t)$ that takes a supercapacitor from a charged state at applied potential $U_{ch} > 0$ V to the fully discharged state at $U = 0$ V. It has been suggested[41] that the optimal linear discharging sweep is as fast as possible, i.e., via a step to $U = 0$ V. This is different from optimal charging with a gentle voltage sweep. For step discharging the ions diffuse passively in and out of the pore, which is in contrast to active migration in the bulk of a supercapacitor as driven by the applied potential during charging. Hence, there is no danger of clogging the pore entrance either with co-ions or counterions. However, could discharging be even faster than what is achieved by a step voltage? This may be expected for an EDLC with planar electrodes, which has been successfully modelled for low voltages through an equivalent $RC$ circuit[59] (see also ref. [49]). The dielectric capacitor of such a circuit can be discharged 'instantaneously' by applying a negative delta-like spike (as obtained from the mathematical model, ignoring dissipation, see Supplementary Note S1), suggesting that an analogous sped-up discharging could be achieved for the corresponding double-layer capacitor (note, however, that the applicability of the circuit model to double-layer capacitors is questionable for such a spike). For a nanoporous supercapacitor, it is unclear how a negative applied potential could speed up counterion desorption, as the electric field vanishes inside nanopores[60]. But there might well be pore-entrance effects. For instance, applying a negative potential difference to a nanopore may deplete counterions from the pore entrance area, leaving the counterion desorption akin to an expansion of a gas into a vacuum, which is faster than into a finite density gas. However, applying a too-negative potential difference may clog the pore entrance with co-ions, in the same way as the attracted counterions impede co-ion desorption in the charging process. To discern how these two competing phenomena play out, we have studied the behaviour of our supercapacitors upon discharging

via a voltage-inversion discharging sweep (Fig. 1e):

$$U(t) = \begin{cases} U_{ch} > 0 & \text{if } t \leq 0, \\ U_{inv} + k_{inv}t & \text{if } 0 < \tau_{inv}, \\ 0 & \text{if } t \geq \tau_{inv}, \end{cases} \quad (10)$$

where $k_{inv}$ is the slope with which the voltage approaches $U = 0$ V after applying a voltage inversion of magnitude $U_{inv} < 0$ V at time $t = 0$, and $\tau_{inv} = -U_{inv}/k_{inv}$ is the time at which $U = 0$ V.

*Results of MD simulations*. Figure 5 shows examples of $U(t)$ given by eq. (10) for a few values of $U_{inv}$ and $k_{inv}$, along with the corresponding numbers of co and counterions obtained by MD simulations. We first analyse how this discharging behaviour depends on $k_{inv}$ at fixed $U_{inv}$. When $k_{inv}$ is too small, i.e., when the applied potential reaches $U(t) = 0$ too slowly, the co and counterions greatly overshoot their final values. This early overshoot is followed by a slow relaxation to equilibrium, evidently giving long discharging times (Fig. 5a–c). When $k_{inv}$ is too high, that is, when the application time of the (varying) negative potential difference is too short (small $\tau_{inv} = -U_{inv}/k_{inv}$), then co and counterions 'undershoot' and discharging proceeds similarly as in the case of step voltage (Fig. 5g–i; step voltage not shown, but cf. Fig. 6a). This suggests that for each inversion voltage, $U_{inv}$, there must be a single optimal $k_{inv}$ that minimizes the discharging time (for this $U_{inv}$).

We then optimized discharging over $k_{inv}$ for a set of inversion voltages $U_{inv}$. The optimal $k_{inv}$ was determined as $k_{inv}$ providing the minimum discharging time, defined as the time at which the charge $Q$ vanishes and exhibits only small fluctuations around $Q = 0$ within 5% of the initial charge (note that it is possible that the charge crosses zero and then decays to $Q = 0$ from the other side). Simulation results for fixed $U_{inv}$ are shown in Fig. 6a, where we compare a few voltage-inversion discharging curves with the step-voltage discharging. Similar to the number of co and counterions, also the charge overshoots when the inversion voltage is applied for too long ($k_{inv} = 2.5$ V/ns in Fig. 6a, corresponding to $\tau_{inv} = 1$ ns, see also Fig. 5a–c). Conversely, for high $k_{inv}$, discharging proceeds akin to the charge response to a step-voltage, albeit slightly faster (compare $k_{inv} = 10$ V/ns and Step in Fig. 6a). Remarkably, the optimal $k_{inv}$ ($k_{inv} \approx 5.5$ V/ns in Fig. 6a) provides a few-fold shorter discharging time, as compared to the step voltage (~0.4 and 1.5 ns, respectively).

In Fig. 6b we plot discharging times, calculated at optimal $k_{inv}$ values, as a function of the inversion voltage $U_{inv}$. This figure clearly demonstrates that there is a global minimum in the discharging times, obtained at $U_{inv} \approx -2.5$ V for our model supercapacitor charged at $U_{ch} = 3$ V (it is noteworthy that an optimal pair ($U_{inv}$, $k_{inv}$) depends on $U_{ch}$). For higher $U_{inv}$ and high $k_{inv}$, the inversion voltage induces strong overshooting (Fig. 5e, f), whereas for lower $U_{inv}$ the system 'undershoots' and discharging proceeds similarly as in the case of step voltage (Fig. 5d, e).

We thus conclude that our voltage inversion scheme (eq. (10)), when properly optimized, provides few-fold lower discharging times than a voltage step. Although we have used same-size ions in our simulations, we expect similar behaviour for cation and anions with different sizes. In this case, an optimal ($U_{inv}$, $k_{inv}$) pair might be different for the cathode and the anode and one may need to compromise the speed of discharging at one of the electrodes. In practice, $U_{inv}$ and $k_{inv}$ must be optimized for the entire supercapacitor.

*Experimental results*. With the same supercapacitor based on the novolac-derived porous carbons (Fig. 1a, b), we carried out experiments to validate the MD predictions that voltage inversion

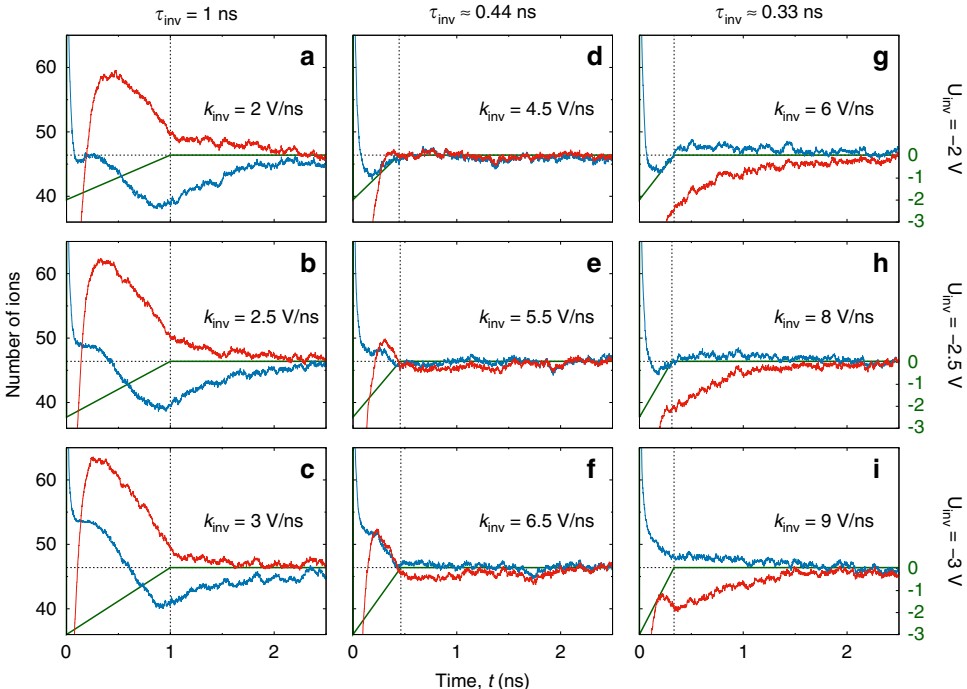

**Fig. 5 Discharging via voltage inversion.** Potential difference and the number of particles for a few values of the inversion voltage $U_{inv}$ (measured with respect to the bulk electrolyte) and slope $k_{inv}$ with which the voltage reverses to approach $U = 0$, see eq. (10). The rows correspond to the same values of $U_{inv}$, as indicated in the plot, and the columns to (approximately) the same values of $\tau_{inv} = -U_{inv}/k_{inv}$ (thin vertical lines). The voltage and the numbers of co and counterions in the pore are shown by green, red, and blue lines, respectively. The thin horizontal lines denote the equilibrium numbers of co and counterions at $U = 0$. The optimal discharging parameters, minimizing the discharging time, are $U_{inv} = -2.5$ V and $\tau_{inv} \approx 0.4$ ($k_{inv} = 5.5$ V/ns, cf., Fig. 6), corresponding to the middle of the nine panels. In all plots the ion diameter $a = 0.5$ nm, the pore width $w = 0.6$ nm and the pore length $\ell = 12$ nm.

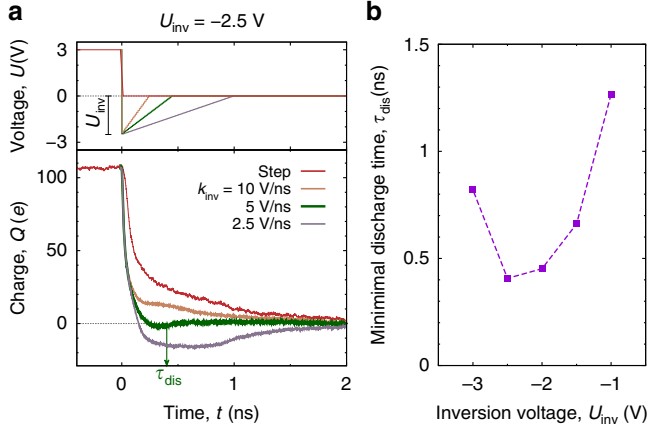

**Fig. 6 Accelerating discharging by voltage inversion. a** Voltage with respect to the bulk electrolyte as a function of time for step-voltage discharging (red line) and for three voltage-inversion sweeps given by eq. (10) (top plot). Bottom plot shows the charge in the nanopore obtained by MD simulations after application of the inversion sweeps, eq. (10), to a model supercapacitor shown in Fig. 1d. The nanopore was initially fully charged at potential $U_{ch} = 3$ V with respect to the bulk electrolyte. The inversion voltage is $U_{inv} = -2.5$ V and the optimal $k_{inv}$ value is $k_{inv} = 5.5$ V/ns. The results are an average of 5 independent simulations. **b** Discharging time as a function of $U_{inv}$, calculated at optimal $k_{inv}$ values. The resulting optimal ($U_{inv}$, $k_{inv}$) pair is ($-2.5$ V, 5.5 V/ns) for our supercapacitor fully charged at $U_{ch} = 3$ V. In all plots the pore width is $w = 0.6$ nm and the pore length $\ell = 12$ nm.

can accelerate discharging. In Fig. 7 we compare a step-voltage discharge ($U_{inv} = 0$ V) with two voltage inversions. We chose $U_{inv} = -2.5$ V based on the MD simulation results (Fig. 6), but the values of $k_{inv}$ for the experimental setup had to be taken much smaller than $k_{inv}$ used for the single nanopore of the MD simulations. Figure 7 demonstrates that the voltage inversions discharge the supercapacitor much faster than the voltage step.

For the voltage inversions, the charge on the electrode becomes negative at intermediate times, after which it becomes positive again. We have not seen this qualitative feature so pronounced in the MD simulations (compare with the green curve in Fig. 6a). One likely has to account for ionic currents in the quasi-neutral macropores of the porous electrodes[58], which are ignored in our MD simulations.

## Discussion

We have investigated how charging and discharging times of supercapacitors can be minimized by judiciously choosing time-depending sweeping rates of potential difference. Previous work has shown by MD simulations that when a step-voltage is applied to an electrode with subnanometre pores, the quickly adsorbed counterions clog the pore entrance and lead to co-ion trapping, causing sluggish charging dynamics[54,55]. Such clogging can be avoided by applying the potential difference slowly via an optimized linear sweep[41]. Here we demonstrated that one may achieve even *faster* charging by adjusting the rate of voltage variation to the rate of co-ion desorption. We presented a general expression for such a non-linear voltage sweep $U(t)$ (via inversion of eq. (2)). Unlike linear sweeps, which require a separate optimization for each potential difference, the proposed $U(t)$ provides

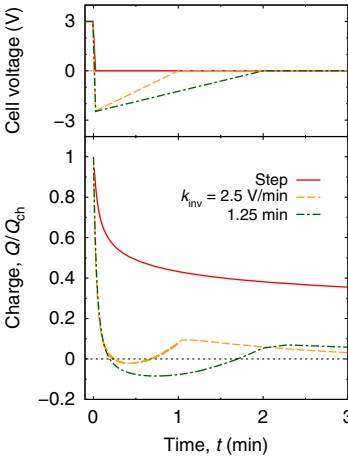

**Fig. 7 Accelerating discharging by voltage inversion for novolac-derived carbons.** Cell voltage as a function of time for step-voltage discharging (red line) and for two voltage-inversion sweeps given by eq. (10) with $U_{inv} = -2.5$ V (top plot). The bottom plot shows the accumulated charge, expressed in terms of the initial charge $Q_{ch}$, after application of the discharging protocols from the top plot. The supercapacitor was initially charged with a step voltage for about 1 h at cell voltage $U_{ch} = 3$ V. For discharging at an elevated temperature and with a lower inversion voltage see Fig. S16.

the optimal charging path for all voltages below $U$. Computing $U(t)$ requires the knowledge of the in-pore counterion density, which is not straightforward to measure. We therefore considered two closed-form approximations, eqs. (6) and (9). These approximate expressions depend on three and two parameters, respectively, which in practical applications can be treated as optimization variables. With MD simulations, we showed that the optimized non-linear sweep can indeed provide a significantly faster charging (Fig. 3). The gain in the charging time over the linear optimal sweep increases for increasing pore length (Fig. 3c), which suggests that the non-linear sweeps can be particularly relevant for realistically long nanopores. We showed experimentally with novolac-derived porous carbons that slow voltage sweeps indeed provide faster charging (Fig. 4), but more work needs to be done to differentiate the linear and non-linear protocols in the regime of full charging.

For discharging we have found that an uncharged state could be reached much faster than diffusively. This might seem surprising given that no electrostatic driving force acts on the in-pore ions. However, applying a 'negative' potential difference to an electrode (i.e., opposite in sign to the potential of a charged state) removes counterions from the pore-entrance, effectively speeding up their desorption while also accelerating the adsorption of bulk co-ions into the pores (Fig. 5). We optimized the discharging time of such a two-parameter voltage inversion procedure, in which the applied potential steps to a negative value and returns to zero with a linear voltage increase (eq. (10)). Using MD simulations, we observed a few-fold decrease in discharging times, as compared to the purely diffusive step-voltage discharging (Fig. 6). We experimentally confirmed these MD findings with the same novolac electrodes. Although we assessed this procedure with subnanometre pores and neat ionic liquids, it shall work for mesoporous electrodes as well as dilute electrolytes, because it is based on temporary speeding up of ion adsorption/ desorption by applying a transient potential difference (Supplementary Note S1C). The finding that discharging can be accelerated may find a useful application in capacitive deionization, where for the production of potable water via ion electrosorption, the operation speed is a very important factor.

## Methods

**Simulations**. We have carried out MD simulations with the simulation package ESPResSo (version 3.3.1)[61–64]. The electrode geometry was built with carbon particles based on a triangular mesh of the slit-shaped surface, resulting in a hexagonal carbon structure. Throughout the study, we used the following parameters: Pore entrance radii of 4 Å, pore closing radii of 2 Å, an accessible pore width of 0.6 nm, and an electrode separation (or bulk size) of 8 nm. We used pore lengths of 12, 16 and 20 nm for our study of the charging behaviour, whereas a single pore length of 12 nm was used in the discharging part.

We used the ICC* algorithm[65] to carry out constant-potential simulations. The ICC* method recalculates the charges on the carbon atoms every time step (2 fs) to model metallic boundary conditions and charge induction caused by ions. This was supplemented by the pre-calculated electrostatic potential $\phi$ (varying in space) due to the voltage applied between the two electrodes; $\phi$ was obtained by solving the Laplace equation[66] of the respective electrode geometry with a reference potential drop (between the electrodes) of 1 V, which was rescaled in accord with a time-dependent potential difference during a simulation.

To gain computational efficiency and consistency with previous studies[41,55,67–71], we used the Weeks–Chandler–Anderson (WCA) potential for the interactions between all particle species. We chose the parameters of this soft-core repulsive interaction as follows: $\sigma_c = 3.37$ Å and $\epsilon_c = 1$ kJ/mol (carbon atoms) and $\sigma = 5$ Å and $\epsilon = 1$ kJ/mol (ions), i.e., the monovalent ions were treated as symmetric WCA particles. The results of ref. [41] show that co-ion trapping and pore clogging occur for both size-symmetric and size-asymmetric ions and hence linear voltage sweeps can accelerate charging in both cases. Likewise, we expect that non-linear sweeps can speed up charging for asymmetric ions but we leave such studies to further work.

We used the velocity-Verlet algorithm[72] in the NVT ensemble to propagate the system and a Langevin thermostat at temperature $T = 400$ K and damping constant $\xi = 10$ ps$^{-1}$. The thermalization dissipates the temperature increase due to Ohmic losses that appear during charging[73,74]. This also alters the dynamics of the system, so the choice of $T$ and $\xi$ will have an impact on the time characteristics such as the charging times presented here. Even though absolute values will depend on the NVT parameters, inter-system comparisons using the same thermostat setup are still valuable. In all simulations of charging, the system was first equilibrated for 4 ns without applied potential before a production run.

## Experiments

*Synthesis of Novolac-derived carbon.* The synthesis of Novolac-derived carbon involves three main steps[57]. Briefly, we first crosslinked our polymer precursor using a solvothermal method. 20 g of Novolac (ALNOVOL PN320, Allnex) was dissolved in 100 mL ethanol. Then, 2.5 g of crosslinker (hexamethylenetetramine) was dissolved in 500 mL mili-Q water. The novolac solution was then added to the crosslinker solution in a 1 L autoclave container. At this step, we observed that the colour of the mixed solution changed from colourless to milky solution indicating the self-emulsion process. Before heating the autoclave, the container was filled with $N_2$ gas to avoid oxidation of our polymer solution. Then, the autoclave was heated to 150 °C with the heating rate of 5 °C/min. The autoclave was held at 150 °C for 8 h and passively cooled down to room temperature. The as-obtained sample was freeze-dried using liquid nitrogen to obtain novolac-beads. As for the second step, we pyrolyzed novolac-beads under an argon atmosphere at 700°C using a heating rate of 2°C/min. The pyrolysis time was 2 h. To enlarge the pore of carbon, as well as to enhance the surface area of carbon, the $CO_2$ activation was necessary. The sample was subjected into the tubular furnace and heated to 1000°C, while feeding $CO_2$ gas with the constant flow rate of 100 cm$^3$/min. The activation time was 2 h. The activated Novolac-derived carbon was named PNC-2h.

*Material characterization.* The electron micrograph of PNC-2h was obtained via field-emission scanning electron microscope (SEM; JEOL-JSM-7500F, JEOL Ltd). Nitrogen gas sorption analysis was performed with a Quantachrome Autosorb iQ system. Before nitrogen sorption, the sample was vacuum degassed at 200 °C for 1 h. After, the sample was heated to 300 °C and hold at such temperature for 20 h at the relative pressure of 0.1 Pa. Yet, the sample are volatile free. We then conduct the nitrogen sorption analysis at the temperature of $-196$ °C using liquid nitrogen. The relative pressure of the nitrogen was increased from $5 \times 10^{-7}$ to 1.0 in 79 steps. A quenched-solid density functional theory (QSDFT) was used to calculate the pore size distribution (PSD) assuming slit-like pores.

The scanning transmission electron microscope experiments were performed in the bight field mode on a JEOL Cs corrected ARM operated at 200 kV equipped with a cold field-emission microscope with a nominal 0.1 nm probe size under standard operating conditions.

*Electrode preparation and cell assembling.* We used free-standing polymer-bound electrode produced as described below[57,75]. The carbon material PNC-2h was first dispersed in ethanol. The mixture was stirred for 5 min, then polytetrafluorethylene (PTFE; 60 mass % in $H_2O$) was added into the carbon-ethanol mixture. The slurry was constantly mixed in the motor until the ethanol was evaporated. The dough-like carbon paste was then pressed into the squarish shape and rolled using the rolling machine to adjust the thickness of the electrode to 120 μm (wet thickness). The carbon electrode was then dried in the vacuum oven at 120 °C overnight. The

resulting electrode consist of 90 mass % carbon and 10 mass % of PTFE. The dry thickness of our working electrode was ca. 100 μm.

We conducted the electrochemistry measurement using our custom-built cell[75]. We used full-cell symmetrical two-electrode setup having PNC-2h electrode as the working and counter electrode. First, PNC-2h electrode was cut in a disk with a diameter of 12 mm. This electrode was attached to a graphite-coated aluminium current collector of the same diameter having the thickness of 37 μm. A 13 mm in diameter glass fibre (GF/F Whatman) was used as separator. After putting all the component into the body of the cell, the cell was tightly closed using spring-loaded titanium piston while leaving hold at the side of the cell open. The cell was then dried in the vacuum oven at 120 °C overnight to remove the residual humidity. Finally, the dried cell was filled by the ionic liquid (EMIM-BF$_4$) in an MBraun Argon-filled glovebox (O$_2$, H$_2$O < 1 p.p.m.).

## Data availability
The data that support the findings of this study are available from the corresponding author upon reasonable request.

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

## Acknowledgements

We thank Jakob Krummacher for fruitful discussions and Cheng Lian for determining the prefactor α from the data of Fig. S9 of the Supplementary Material of ref. [58]. M.J. and S.K. acknowledge support from S. Dietrich (MPI-IS, Stuttgart). V.P. and P.S. thank Eduard Arzt (INM) for his continued support. C.H. and K.B. were funded by Deutsche Forschungsgemeinschaft (DFG, German Research Foundation) under Germany's Excellence Strategy - EXC 2075 - 390740016 and through Project Number 327154368 – SFB 1313.

## Author contributions

S.K., C.H. and K.B. initiated the research. K.B. performed the simulations. V.P. designed and supervised the experimental work, P.S. performed the experiments and B.L.M. did the TEM analysis. S.K. and M.J. derived the equations and drafted the manuscript. All authors contributed to the discussion of the results and editing of the manuscript.

## Funding

## Competing interests

The authors declare no competing interests.
