## [Peer Review File · Nature Communications]

REVIEWER COMMENTS

Reviewer #1 (Remarks to the Author):

The manuscript by Breitsprecher et al. reports very interesting and important findings regarding how to optimize the charging/discharging rate in electric double layer capacitors (EDLCs) that utilize nanoporous electrodes. These energy storage devices attracted quite a bit of attention as potential alternative for Li-ion batteries. Significant progress has been made over last decade in designing nanostructured electrodes that maximize electrolyte accessible surface area for charge storage. While this allowed to significantly increase the capacitance of EDLCs, it also brought additional challenges with limited charging rates. Because during charging/discharging processes ions have to diffuse in/out of the nanopores, the charging rate have to be carefully tuned to optimize the coupling between ion transport and charge storage mechanisms. This manuscript focuses on this challenging issue which was previously overlooked. The authors clearly demonstrate that by optimizing the voltage sweep a significantly faster (compared to standard step or linear voltage change) charging/discharging rates can be achieved.

The manuscript provides theoretical foundation for proposed hypothesis, demonstrates conceptual proof by conducting coarse-grained molecular dynamics simulations, and then confirms the predictions by experiments. Overall this is an excellent manuscript with a very significant and transformative findings that will likely be impactful in many areas beyond just the energy storage devices. Therefore I strongly support publication of this manuscript.

I only have couple minor questions/comments:

- 1) Both simulations and experiments were conducted for very narrow pores (the width is comparable to ion dimensions). Would similar optimization approach be applicable to wider pores, e.g. pores with width equal 2-4 ion diameters?
- 2) How the adsorption/desorption times shown in Fig. 3b were defined? It might be useful to indicate with arrows on Figure 3a for $\tau=12$ case.
- 3) Authors showed that for each inversion voltage there is an optimal slope for voltage reversal to zero (k_{inv}). Is there any way to predict the optimal k_{inv} for a given U_{inv} (or even optimal k_{inv} , U_{inv} combination) based on known characteristics of electrolyte and electrode? I understand that this question cannot be answered within the scope of this manuscript, but if authors have any thoughts it might be useful to put a brief discussion.

Reviewer #2 (Remarks to the Author):

The paper from Breitsprecher et al. is from a well-known group of authors led by S. Kondrat who did several key papers in the field the past years. It deals with an interesting and hot topic, which is the understanding of the kinetics of ion fluxes in carbon nanopores for capacitive charge storage applications. The objective is to answer the question: is it possible to design strategies which can improve the charge (and discharge) kinetics and efficiency of supercapacitor electrodes. The paper thus addresses an important topic. There are interesting results in it (modelling), but also some concerns in other places (experiments) so that it is difficult to make a final decision at this stage. The paper should be revised according to the following comments and questions.

The first part of the paper deals with the optimization of the charge procedure, by using non-linear voltammetry profile.

- 1) A first question deals with the theoretical approach. Figure 3a shows the change of the potential, charge and ion number vs the charging time. It shows that a non-linear profile is better than the "optimal" linear potential sweep. One concern is that with the optimal linear sweep, the

charging time is different compared to the non-linear and this makes difficult the comparison between the two strategies. It would have been more convincing to show the change of the U, Q and ion number for a linear sweep during a total time of 12 ns, to compare with the non-linear profile showing the best performance. If results are still different, then some conclusions could be made. In the current situation (linear sweep achieved during a time > 12 ns), no conclusions can be formally drawn.

2) Experimental part:

A conclusion of the experimental part is that the results "are inconclusive regarding the comparison of the linear and non-linear sweep". This is clearly shown in Figure 4. Authors explained that this is because the optimum sweep rate is below 0.1 mV per s. This is certainly right. But now, what could be the reason why?

Figure 4 brings a part of the answer: the ohmic drops. This was more or less predictable since the authors used a neat ionic liquid (EMIM,BF₄) with limited ionic conductivity, as most of theoreticians use for modelling (not a criticism). Indeed, Figure 4 shows that the charge still increases after the potential limit has been reached, which can be attributed to a kinetics controlled by ohmic drops. The more the current, the more the IR. Then, charge continue to increase as a result optimization of the potential distribution inside the electrodes (current decreases). The high resistance of the cell and the electrodes can also be seen from the time constant needed to charge the device (several minutes). It can also explain the presence of a second exponential in the equation (bottom of page 10). And it explains why the optimum scan rate is so low (low scan rate means low capacitive current) ...

Now, an important concern is that the electrode (and cell) resistance is not taken into account into the model. As a result, it difficult – or impossible – to compare the set of experimental data and the results from the modelling approach since the model does not take into account the resistive effects. One way to tackle this issue is to run experiments in a solvent + salt electrolyte (acetonitrile + EMI,BF₄ for instance) with $\sim \times 10$ ionic conductivity, resulting in a drastic decrease of the ohmic drop. Then, I would suggest to redo the experiments in such electrolyte with higher ionic conductivity to decrease the electrolyte bulk resistance and the resistance of the electrolyte confined in the pores of the carbon electrode.

On the same line, it is normal that the non-linear sweep gives better results than the step-voltage approach, because in the latter the current "I" flowing in the cell is maximum when switching the potential to 0, resulting in a maximum ohmic drop "IR". Then, it would be interesting to compare the non-linear sweep to the potential step approach by using smaller potential steps: moving from 3 V down to 1.5 V, then 1.5 V down to 0 V for instance. Would the results be the same? I guess no: the difference between the 2 approaches should be smaller. But this is important to check.

Discharge process:

Same comments apply to the discharge part. The modeling part is nicely done and shows interesting results. However, the experimental data in Figure 7 may simply reflect the limitation by ohmic drops (uneven potential distribution inside the electrodes). I would then recommend the authors to redo the experimental part by using more conducting electrolyte such as acetonitrile + 2M EMI,BF₄ for instance.

Minor comments: authors mention in several places in the manuscript that a step-voltage strategy is worst than linear sweep to charge the electrodes. This is only true in their approach, when considering a very large step voltage (3 V). One has to keep in mind that current potentiostats achieve CV experiments using incremental step voltage of few μ V of tens of μ V. So, authors should be more precise and mention that this is true for their specific case (large step voltage).

In summary, the paper contains interesting results from the modelling point of view. However, the experimental part has to be revised. More specifically, authors should use a cell design that allows for limiting the ohmic drops in the electrodes and cell, by using a solvent + salt electrolyte. Such an approach will make the comparison between the model and experimental results more accurate, offering the possibility to assess the quality of the theoretical approach.

Reviewer #3 (Remarks to the Author):

This manuscript investigates the charging/discharging times of capacitors consisting of a room temperature ionic liquid (RTIL) and nanoporous electrodes, and more precisely the possibility to optimize these charging/discharging times using time-dependent voltages more complex than a step function (i.e. switching on/off). The study combines analytical calculations under strong but reasonable simplifying assumptions, molecular simulations with model slit pores and experimental results. Even though the theoretical considerations do not readily apply to the real materials used for the experiments, the latter support the general idea of optimizing the voltage sweep, in particular with a voltage inversion followed by a linear decay to achieve a fast discharge. Even though this work builds substantially upon previous work by some of the authors (Ref. 42), it also provides new theoretical contributions and, importantly, an experimental counterpart. The ideas developed by the authors could be useful in several practical contexts involving capacitors with nanoporous electrodes, including energy storage, capacitive deionization and electrochemical heat harvesting.

I may eventually recommend publication if the authors address the points listed below.

1) A first important issue is to clarify the position with respect to Ref. 42 by some of the authors, which included molecular dynamics simulations of the same slit-pore capacitor and related analysis of the ionic diffusion and charging times. As mentioned above, the present work does include new theoretical contributions and an experimental counterpart. While the previous work is correctly referenced, it is important to emphasize the novel aspects introduced in the present one, in particular with respect to some ideas on optimization already present in Ref. 42.

2) Similarly, the authors should discuss more the new aspects of the present work with respect to Ref. 51 by other authors (now published as *Phys. Rev. Lett.* 124, 076001, 2020). At present the manuscript compares the MD results with some theoretical results of that reference, which to the best of my knowledge considers electrolytes, and the applicability to RTILs, while relevant, should be discussed. The analogy between diffusive charging and a RC circuit / transmission line model was already present in this reference (and others cited), and used to analyze the scaling of charging time. As for the previous comment, such a discussion would help emphasizing the novelty of the present work.

3) While it may not have been already considered for supercapacitors, the idea of optimizing a time-dependent protocol to drive a system from an initial to a target state in a minimal time is not new and has already been exploited/illustrated in many contexts, such as pulse shaping in NMR or lasers, or the concepts of shortcut to adiabaticity for quantum systems (*Adv. At. Mol. Opt. Phys.* 62, 117–169, 2013) and engineered swift equilibration with experiments on optical tweezers on colloids (see e.g. *Nature Physics*, 12, 843, 2016) or an AFM tip (*Appl. Phys. Lett.* 109, 113502, 2016). The introduction and/or discussion should place the present work in this broader context.

4) While the analytical/simulation and experimental parts are very complementary, they also deal with different objects. The real materials (novolac-derived carbons) are very different from the single slit pores considered in the theoretical counterpart. It is therefore necessary to discuss the effects of pore geometry and material heterogeneity on the extrapolation of the model for the single slit pore, as well as the role of the bulk region between the electrodes (alluded to on pages 11 and 16) — not present in the molecular simulation setup, but that could play an important role in the experiments. In addition, even though the PSD of the novolac-derived carbon indicates a relatively narrow size distribution, some data/reference should be provided to clarify whether the geometry of the pores of this material resembles the simpler one considered in the theoretical part.

5) In the derivation of the non-linear sweep function of page 5, an important yet (unless I missed it) not explicit assumption is that K , the number of « imaginary layers », which quantifies the number of ions in the pore at a given time, is assumed to be a function only of the instantaneous voltage U . However, the purpose of this work is to show that the protocol $U(t)$ can be optimized to achieve the fastest charge/discharge. Based on the arguments developed in the manuscript that this is possible because the dynamics within the pore depends on its instantaneous composition, it seems that in principle several pore states could be obtained for a given instantaneous voltage U_0 , depending on the previous history of the system from the initial time and under the protocol $U(t)$ up to the current value U_0 . The authors should clarify the assumptions on $U(t)$ and/or the dynamics of the system underlying the analysis, which assumes that the state only depends on the voltage at a given time (and hence that the instantaneous pore state depends only on the instantaneous voltage).

6) The analysis of the MD simulations includes a discussion of adsorption/desorption times (presented in Figure 3b), but these times are never defined.

7) On page 10, it is said that « The data suggest a linear increase of t_{charge} with l in the latter case, consistent with the asymptotic analysis below eq. (6). In the former case $t_{\text{charge}} \sim l^2$ ». However that the discussion below Eq. (6) applies in specific limits: are they the relevant ones here?

8) On page 11, the authors compare their experimental results with the theoretical predictions of Ref. 51 for the slow time scale, which involves « a salt-concentration dependent factor (of order unity for dilute electrolytes) ». Which value was considered here, since RTILs are not dilute electrolytes?

9) On page 13, the authors say to motivate this part that « for discharging, there is no danger of clogging the pore entrance either with co-ions or counter-ions. » Could the authors explain why?

10) The MD results for the discharge (in particular Figure 5) includes a discussion of the number of co- and counter-ions inside the electrodes. The ions are identical (only with opposite charges) in MD, but this is not the case in real systems, in particular EMIM-BF₄ in the present experiments. Could the authors comment on the effect of this asymmetry on their optimization procedure? Of course the exact result will be system specific, but it is probably possible to make some general comments (or at least underline the limitations of the present case).

11) The caption of Figure 6 refers to the optimal $(U_{\text{inv}}, k_{\text{inv}})$ pair, but based on the rest of their discussion presumably this optimum depends on the initial charging state, hence U_{ch} . Could the authors discuss this point?

12) At the bottom of page 15, the authors introduce « the minimum discharging time, defined as the time at which the charge Q vanishes and exhibits only small fluctuations around $Q = 0$ ». How is it determined in practice? Is it only a visual estimate, or is there a more systematic criterion (or algorithm) to measure it from the data?

13) On page 16, it is said that « the values of k_{inv} for the experimental setup had to be taken much larger than k_{inv} used for the single nanopore of the MD simulations. » Isn't it much smaller instead (i.e. much longer timescale)?

14) On page 18, the authors indicate that « work needs to be done to differentiate the linear and non-linear protocols in the regime of full charging. » It would indeed be relevant to discuss (already in the present work) some other considerations than the charging/discharging time, in particular the work and/or power (this last aspect is only mentioned in passing in the Supplementary Material, at the end of section S1C for the RC circuit).

15) At the very end of the conclusion, the authors indicate that: « The finding that discharging can be accelerated may find a useful application in capacitive deionization, where for the production of potable water via ion electrosorption, the operation speed is a very important factor. » While this may be true, the electrolyte in this context is very different from the present case (dilute electrolyte solution vs RTILs). It would be interesting to add a comment on how the present findings may hold (or not) for dilute electrolytes, or at the very least necessary to include a caveat on the applicability to these systems.

Response to the Referee reports

We thank all Referees for carefully reading our manuscript and for helpful comments and suggestions, which we have all taken into account. The detailed response to the Referee comments is provided below. In our response, the Referee comments are in italic. We provide the amended text when necessary or refer to the pages in the revised manuscript, where the changes have been made in response to the Referee comment. We also provide a diff file, wherein all changes are highlighted.

Response to Reviewer #1

Referee Comment: The manuscript by Breitsprecher et al. reports very interesting and important findings regarding how to optimize the charging/discharging rate in electric double layer capacitors (EDLCs) that utilize nanoporous electrodes. These energy storage devices attracted quite a bit of attention as potential alternatives for Li-ion batteries. Significant progress has been made over last decade in designing nanostructured electrodes that maximize electrolyte accessible surface area for charge storage. While this allowed to significantly increase the capacitance of EDLCs, it also brought additional challenges with limited charging rates. Because during charging/discharging processes ions have to diffuse in/out of the nanopores, the charging rate have to be carefully tuned to optimize the coupling between ion transport and charge storage mechanisms. This manuscript focuses on this challenging issue which was previously overlooked. The authors clearly demonstrate that by optimizing the voltage sweep a significantly faster (compared to standard step or linear voltage change) charging/discharging rates can be achieved.

*The manuscript provides theoretical foundation for proposed hypothesis, demonstrates conceptual proof by conducting coarse-grained molecular dynamics simulations, and then confirms the predictions by experiments. Overall this is an excellent manuscript with a very significant and transformative findings that will likely be impactful in many areas beyond just the energy storage devices. **Therefore I strongly support publication of this manuscript.***

Reply: We thank the Referee for the positive assessment of our work.

Referee Comment: I only have couple minor questions/comments:

1) Both simulations and experiments were conducted for very narrow pores (the width is comparable to ion dimensions). Would similar optimization approach be applicable to wider pores, e.g. pores with width equal 2-4 ion diameters?

Reply: For charging we have an approximate but general expression, eq (2), which relates equilibrium properties with an optimal charging protocol via $K(u)$. It is therefore possible to obtain an optimal $U(t)$ for arbitrary pore geometry and structure, provided the pore length

(~carbon particle radius) and the in-pore counterion density are known. This shall in principle work also for pore widths 2-4 ion diameters; the absence/presence of trapping and its degree enters into the sweep function via $K(U)$ dependence. How well this approximation works for wider pores remains to be seen. However, we note that in the extreme case of mesopores, an “effective pore length” in eqs (2) and (5) would be the thickness of the double layer. Then the ‘time scale’ (duration) of voltage application would become extremely short (see eq (5)), effectively leading to a step potential.

It will be really interesting to investigate a “transition regime” and the dependence on the pore width, but it is beyond the scope of the present work.

For discharging, the voltage-inversion procedure does not need to avoid ion clogging or congestions. Rather, since step-voltage discharge is essentially diffusive, this protocol aims to accelerate the removal of counter-ions and to speed up the adsorption of co-ions. Thus, this method ought to work for any type of electrodes, including flat electrodes (as we show in SI, Section S1C).

Motivated by the Referee questions, we now emphasize this in the manuscript (page 19):

Although we assessed this procedure with subnanometre pores and neat ionic liquids, it shall work for mesoporous electrodes as well as dilute electrolytes, because it is based on temporary speeding-up of ion adsorption/desorption by applying a transient potential difference.

We thank the Referee for this question.

Referee Comment: 2) How the adsorption/desorption times shown in Fig. 3b were defined? It might be useful to indicate with arrows on Figure 3a for $\tau=12$ case.

Reply: We have added the definition of adsorption/desorption times (page 10):

To determine the optimal τ , we studied the τ dependence of the co-ion desorption and counter-ion adsorption times, which are the times needed by the co-ion and counter-ion densities to reach their equilibrium values corresponding to the final applied voltage (these times are not the same [42] and depend on charging protocol $U(t)$; see Figs. S10 and S11).

We also added two Supplementary plots (S10 and S11) on which we indicate these times by arrows/lines.

We thank the Referee for spotting this.

Referee Comment: 3) Authors showed that for each inversion voltage there is an optimal slope for voltage reversal to zero (k_{inv}). Is there any way to predict the optimal k_{inv} for a given U_{inv} (or even optimal k_{inv} , U_{inv} combination) based on known characteristics of electrolyte and electrode? I understand that this question cannot be answered within the

scope of this manuscript, but if authors have any thoughts it might be useful to put a brief discussion.

Reply: This is a very good question. We had in fact discussed such possibilities before submitting the original version of the manuscript. Developing such a theory is not a trivial task and, unfortunately, at present it is not clear to us how to do it in an elegant way, avoiding numerical calculations or simulations.

Response to Reviewer #2

Referee Comment: The paper from Breitsprecher et al. is from a well-known group of authors led by S. Kondrat who did several key papers in the field the past years. It deals with an interesting and hot topic, which is the understanding of the kinetics of ion fluxes in carbon nanopores for capacitive charge storage applications. The objective is to answer the question: is it possible to design strategies which can improve the charge (and discharge) kinetics and efficiency of supercapacitor electrodes. The paper thus addresses an important topic. There are interesting results in it (modelling), but also some concerns in other places (experiments) so that it is difficult to make a final decision at this stage. The paper should be revised according to the following comments and questions.

Reply: We thank the Referee for the positive assessment of our work and for his/her insightful comments, which we have taken into account. Motivated by his/her comments, we have performed additional experiments. We hope that our work is now suitable for publication in Nature Communications.

Referee Comment: The first part of the paper deals with the optimization of the charge procedure, by using non-linear voltammetry profile.

1) A first question deals with the theoretical approach. Figure 3a shows the change of the potential, charge and ion number vs the charging time. It shows that a non-linear profile is better than the “optimal” linear potential sweep. One concern is that with the optimal linear sweep, the charging time is different compared to the non-linear and this makes difficult the comparison between the two strategies. It would have been more convincing to show the change of the U , Q and ion number for a linear sweep during a total time of 12 ns, to compare with the non-linear profile showing the best performance. If results are still different, then some conclusions could be made. In the current situation (linear sweep achieved during a time > 12 ns), no conclusions can be formally drawn.

Reply: We have developed the linear and non-linear charging protocols in order to minimize the charging time. In Fig. 3a, the linear sweep charging is *optimized* in the sense that any slower or faster sweep rate increases the charging time. In other words, there is *no* linear sweep that provides charging time 12ns. Sweeping rates that are faster than the optimal one lead to co-ion trapping and hence slower charging.

However, to answer the Referee’s question more thoroughly, we have performed additional simulations with the sweep rate 0.5 V/ns that *stops* at 12 ns. The results are presented in new **Supplementary Figure S9**, which shows that more co-ion trapping occurs for the linear as compared to the non-linear charging protocol. We emphasize now in the caption to Fig. 3 that we present the linear protocol that is optimized and refer to **Supplementary Figure S9** for an additional comparison.

Referee Comment: 2) Experimental part:

A conclusion of the experimental part is that the results “are inconclusive regarding the comparison of the linear and non-linear sweep”. This is clearly shown in Figure 4. Authors explained that this is because the optimum sweep rate is below 0.1 mV per s. This is certainly right. But now, what could be the reason why?

Figure 4 brings a part of the answer: the ohmic drops. This was more or less predictable since the authors used a neat ionic liquid (EMIM,BF₄) with limited ionic conductivity, as most of theoreticians use for modelling (not a criticism). Indeed, Figure 4 shows that the charge still increases after the potential limit has been reached, which can be attributed to a kinetics controlled by ohmic drops. The more the current, the more the IR. Then, charge continue to increase as a result optimization of the potential distribution inside the electrodes (current decreases). The high resistance of the cell and the electrodes can also be seen from the time constant needed to charge the device (several minutes). It can also explain the presence of a second exponential in the equation (bottom of page 10). And it explains why the optimum scan rate is so low (low scan rate means low capacitive current) ...

Now, an important concern is that the electrode (and cell) resistance is not taken into account into the model. As a result, it difficult – or impossible – to compare the set of experimental data and the results from the modelling approach since the model does not take into account the resistive effects. One way to tackle this issue is to run experiments in a solvent + salt electrolyte (acetonitrile + EMI,BF₄ for instance) with ~ x10 ionic conductivity, resulting in a drastic decrease of the ohmic drop. Then, I would suggest to redo the experiments in such electrolyte with higher ionic conductivity to decrease the electrolyte bulk resistance and the resistance of the electrolyte confined in the pores of the carbon electrode.

Discharge process:

Same comments apply to the discharge part. The modeling part is nicely done and shows interesting results. However, the experimental data in Figure 7 may simply reflect the limitation by ohmic drops (uneven potential distribution inside the electrodes). I would then recommend the authors to redo the experimental part by using more conducting electrolyte such as acetonitrile + 2M EMI,BF₄ for instance.

Reply: The Referee has raised interesting questions.

The total resistance in the simulation model comes from both the (two) narrow pores *and* from the bulk. We agree with the Referee that the contribution of bulk and electrodes might be different in the experimental and simulation systems. The results below suggest however that the major contribution in both systems is due to ultra narrow pores, which hinder fast charging, rather than from the bulk.

We have followed the Referee suggestion and performed experiments with 2M EMIM,BF₄ in acetonitrile. The result is shown below:

[redacted]

However, motivated by the Referee's question and to address it in the manuscript for the benefit of potential readers, we have additionally performed experiments with neat EMIM-BF₄ at an elevated temperature, which also provides higher conductivity. These results are consistent with the previous results at room temperature and are included as Supplementary plots to the MS (**Figures S12, S13 and S16**).

Referee Comment: On the same line, it is normal that the non-linear sweep gives better results than the step-voltage approach, because in the latter the current “*I*” flowing in the cell is maximum when switching the potential to 0, resulting in a maximum ohmic drop “*IR*”. Then, it would be interesting to compare the non-linear sweep to the potential step approach by using smaller potential steps: moving from 3 V down to 1.5 V, then 1.5 V down to 0 V for instance. Would the results be the same? I guess no: the difference between the 2 approaches should be smaller. But this is important to check.

Reply: We are not sure we clearly understand this Referee’s question. It seems like the Referee is talking about charging, yet she/he suggests a discharging protocol.

Assuming the Referee means charging, we do not share the intuition that nonlinear charging would naturally yield a faster charge buildup due to decreased ohmic drop as compared to a step voltage. From our RC analogy, we see that a voltage step charges fastest. However, this also means that it has the highest average current, meaning it has the largest average voltage drop over the resistor. In other words, in the RC circuit, having the largest voltage drop does not conflict with having the shortest charging time. Our supercapacitor model does not adhere to this logic.

Considering the in-between step at 1.5V, we do not think it would help either the charging or the discharging process. The non-linear protocol was designed in such a way as to provide the optimal charging path. Charging first to 1.5 volts would still lead to some co-ion trapping that would hinder fast charging.

For discharging, it has been shown in Ref. [42] that discharging with a step to 0 does better than any linear sweep with a finite sweeping rate. The intermediate step at 1.5V would just be a stepwise approximation to such a worse-performing linear sweep.

Referee Comment: Minor comments: authors mention in several places in the manuscript that a step-voltage strategy is worst than linear sweep to charge the electrodes. This is only true in their approach, when considering a very large step voltage (3 V). One has to keep in mind that current potentiostats achieve CV experiments using incremental step voltage of few μV of tens of μV . So, authors should be more precise and mention that this is true for their specific case (large step voltage).

Reply: We have used 3V to compare the step-voltage and the linear and non-linear sweep chargings because at this applied potential difference co-ion trapping occurs for short pores. We have performed additional simulations that show that for longer pores co-ion trapping occurs at lower voltages (**Supplementary Figure S4**). However, such pores are computationally much more expensive to simulate. We now comment on this in the manuscript (**page 10**):

In all simulations, we chose a large potential difference $U = 3 \text{ V}$. The reason is that, at this U , co-ion trapping, which we aim to circumvent with slow voltage sweeps, occurs for relatively short, computationally feasible pores. However, for longer pores the trapping and pore clogging occur at lower potential differences (Fig. S4), suggesting that our approach applies to a wider voltage range.

Equation (2) is a general, albeit approximate, expression that links the equilibrium properties with the speed of optimal charging via $K(u)$. Unlike linear sweeps, which require a separate optimization for each working voltage, eq (2) is valid at any applied potentials and provides the fastest charging protocol for all voltages below U . At a low voltage, this protocol amounts to applying the voltage very quickly, nearly in a step-like manner, as Fig. 2b shows. We have clarified this issue in the manuscript (**pages 5 and 19**):

Equation (2) is an implicit equation for the time-dependent sweep function $U(t)$ in terms of $K(U)$. This equation provides the optimal sweep function for charging to an arbitrary po-

tential difference below U .

and

Unlike linear sweeps, which require a separate optimization for each potential difference, the proposed $U(t)$ provides the optimal charging path for all voltages below U .

Referee Comment: In summary, the paper contains interesting results from the modelling point of view. However, the experimental part has to be revised. More specifically, authors should use a cell design that allows for limiting the ohmic drops in the electrodes and cell, by using a solvent + salt electrolyte. Such an approach will make the comparison between the model and experimental results more accurate, offering the possibility to assess the quality of the theoretical approach.

Reply: We thank the Referee for his/her insightful comments which have helped us improve the manuscript significantly. We hope that our work is now suitable for Nature Communications.

Response to Reviewer #3

Referee Comment: This manuscript investigates the charging/discharging times of capacitors consisting of a room temperature ionic liquid (RTIL) and nanoporous electrodes, and more precisely the possibility to optimize these charging/discharging times using time-dependent voltages more complex than a step function (i.e. switching on/off). The study combines analytical calculations under strong but reasonable simplifying assumptions, molecular simulations with model slit pores and experimental results. Even though the theoretical considerations do not readily apply to the real materials used for the experiments, the latter support the general idea of optimizing the voltage sweep, in particular with a voltage inversion followed by a linear decay to achieve a fast discharge. Even though this work builds substantially upon previous work by some of the authors (Ref. 42), it also provides new theoretical contributions and, importantly, an experimental counterpart. The ideas developed by the authors could be useful in several practical contexts involving capacitors with nanoporous electrodes, including energy storage, capacitive deionization and electrochemical heat harvesting.

I may eventually recommend publication if the authors address the points listed below.

Reply: We thank the Referee for the careful reading of the manuscript and stimulating suggestions, which we have all taken into account.

Referee Comment: 1) A first important issue is to clarify the position with respect to Ref. 42 by some of the authors, which included molecular dynamics simulations of the same slit-pore capacitor and related analysis of the ionic diffusion and charging times. As mentioned above, the present work does include new theoretical contributions and an experimental counterpart. While the previous work is correctly referenced, it is important to emphasize the novel aspects introduced in the present one, in particular with respect to some ideas on optimization already present in Ref. 42.

Reply: We wrote in the Abstract that “In an earlier work we showed for a simple model that a slow voltage sweep charges ultra-narrow pores quicker than an abrupt voltage step” and continue that “Herein, we verify this finding experimentally”. This is an important point, given the simplicity of the model and the complexity of the real system.

We then emphasize the novelty saying that “we develop a non-linear voltage sweep” which “can charge a nanopore even faster than the corresponding optimized linear sweep”.

We elaborate on this even more in the introduction. We refer to Ref [42] which showed that “Pore clogging can be avoided by applying the potential difference with a linear sweep, that is, by varying it with a constant rate”. We emphasize the novelty saying “We show in this article that charging can be made even faster than the fastest (optimized) linear sweep, if the

variation of the applied potential is matched to the actual rate of co-ion desorption. We propose a general expression for a non-linear sweep function...”.

In our opinion, these are very honest and clear statements. We state now additionally that linear sweeps were developed in Ref. 42 to make the novelty even more clear.

However, we agree with the Referee that the novelty of this work was not emphasized regarding discharging. We have added a new paragraph in Introduction contrasting this aspect of the present work with Ref [42] (page 4):

Finally, for discharging, ref. [42] found that a step voltage unloads a supercapacitor faster than any finite-rate linear potential sweep. Here we demonstrate, with MD simulations and experiments, that a supercapacitor can discharge even faster if we apply a non-linear sweep consisting of a voltage inversion followed by a linear sweep to zero.

Referee Comment: 2) Similarly, the authors should discuss more the new aspects of the present work with respect to Ref. 51 by other authors (now published as *Phys. Rev. Lett.* 124, 076001, 2020). At present the manuscript compares the MD results with some theoretical results of that reference, which to the best of my knowledge considers electrolytes, and the applicability to RTILs, while relevant, should be discussed. The analogy between diffusive charging and a RC circuit / transmission line model was already present in this reference (and others cited), and used to analyze the scaling of charging time. As for the previous comment, such a discussion would help emphasizing the novelty of the present work.

Reply: The authors of Ref. 51 (now Ref. 56) indeed mainly looked at dilute electrolytes but also shortly discussed a model RTIL between two electrodes (i.e. not accounting for porosity through their parameter n) on p. 7,8 of their Supplemental Material. There, it was shown qualitatively that their model RTIL also exhibited a two-step relaxation on two broadly-separated timescales. As for the novel features of our work compared to Ref.56: While both works deal with the charging behavior of supercapacitors, approaches differ greatly: Our current manuscript stands in a tradition of studying supercapacitor charging through MD simulations, scrutinizing the nanoscale dynamics of finite-size ions entering narrow pores. As computational resources limit such studies to setups of several nanometers at most (Fig. 1d), charge relaxation also happens in nanoseconds (Fig. 3a). Conversely, in supercapacitor experiments, charging times are typically minutes (Fig. 4). To bridge this 12 orders of magnitude gap in charging times, Ref. 56 proposed a model that accounts for both microscopic charging and ionic fluxes over macroscopic length scales. Such a multiscale study, however, can never describe relevant nanoscale ion dynamics in as much detail as we did here. In fact, the model of Ref. 56 is one dimensional (ionic currents flow perpendicularly through permeable metallic sheets that mimic porous electrodes but do not truly model their morphology) and mostly ignores ionic size effects (except in the aforementioned Supplemental Material). Hence, nanoscale pore clogging---which, as we have shown, can dramatically influence even (macroscopic) supercapacitor charging experiments---is out of reach for the model of Ref. 56 in its current form. Hence, our current

manuscript and Ref. 56 give complementary information on supercapacitor charging. We now emphasize this point on page 12:

Conversely, the model of ref. [56] does not describe the dynamics of finite size ions in nanopores in as much detail as our simulations do, and hence, ref. [56] could not account for the pore clogging effects central to our article.

Referee Comment: 3) While it may not have been already considered for supercapacitors, the idea of optimizing a time-dependent protocol to drive a system from an initial to a target state in a minimal time is not new and has already been exploited/illustrated in many contexts, such as pulse shaping in NMR or lasers, or the concepts of shortcut to adiabaticity for quantum systems (*Adv. At. Mol. Opt. Phys.* 62, 117–169, 2013) and engineered swift equilibration with experiments on optical tweezers on colloids (see e.g. *Nature Physics*, 12, 843, 2016) or an AFM tip (*Appl. Phys. Lett.* 109, 113502, 2016). The introduction and/or discussion should place the present work in this broader context.

Reply: We thank the Referee for pointing this out. We have amended the text in the MS appropriately (page 4):

In other contexts, optimization of time-dependent protocols driving a system from an initial to a final state has been investigated, for instance, to find shortcuts to adiabaticity in quantum systems [48, 49], to engineer swift equilibration of Brownian particles [50] and AFM tips [51], etc.

Referee Comment: 4) While the analytical/simulation and experimental parts are very complementary, they also deal with different objects. The real materials (novolac-derived carbons) are very different from the single slit pores considered in the theoretical counterpart. It is therefore necessary to discuss the effects of pore geometry and material heterogeneity on the extrapolation of the model for the single slit pore, as well as the role of the bulk region between the electrodes (alluded to on pages 11 and 16) — not present in the molecular simulation setup, but that could play an important role in the experiments. In addition, even though the PSD of the novolac-derived carbon indicates a relatively narrow size distribution, some data/reference should be provided to clarify whether the geometry of the pores of this material resembles the simpler one considered in the theoretical part.

Reply: With the state-of-the-art simulation techniques and available computational resources, it is essentially impossible to match experiments and simulations quantitatively. Our choice of the slit pore model is based on the QSDFT data, which indicates that our novolac-derived carbons consist mainly of slit pores. We now include this result in the manuscript (Fig. S3 in the SI) and comment on page 4:

We assess the benefits of such a non-linear voltage sweep in MD simulations of a model supercapacitor and in experiments with novolac-derived carbon electrodes featuring a narrow pore-size distribution and mainly slit-shaped pores (Fig. 1 and Fig. S3).

To get a deeper physical insight into the charging dynamics of our real material, we have therefore chosen to use only slit pores, as this allows the clearest interpretation of physical mechanisms. However, we think that co-ion trapping and pore clogging occur generally in narrow pores. Nevertheless, it will be interesting to perform similar analyses for other pore geometries and particularly for amorphous carbon materials as in Ref. [21,30]. We leave such studies for future work.

Referee Comment: 5) In the derivation of the non-linear sweep function of page 5, an important yet (unless I missed it) not explicit assumption is that K , the number of « imaginary layers », which quantifies the number of ions in the pore at a given time, is assumed to be a function only of the instantaneous voltage U . However, the purpose of this work is to show that the protocol $U(t)$ can be optimized to achieve the fastest charge/discharge. Based on the arguments developed in the manuscript that this is possible because the dynamics within the pore depends on its instantaneous composition, it seems that in principle several pore states could be obtained for a given instantaneous voltage U_0 , depending on the previous history of the system from the initial time and under the protocol $U(t)$ up to the current value U_0 . The authors should clarify the assumptions on $U(t)$ and/or the dynamics of the system underlying the analysis, which assumes that the state only depends on the voltage at a given time (and hence that the instantaneous pore state depends only on the instantaneous voltage).

Reply: Yes indeed, it is possible to obtain several states at time t depending on $U(t)$. However, the charging protocol is designed in such a way that for the *optimal* sweep function the number of co-ions at time t corresponds to the equilibrium value at the current value of the voltage. This is done by charging just as slowly as necessary, to allow the co-ions to reach this equilibrium value, but not slower (in order to minimize the charging time). We have clarified this important point in the MS (page 8):

We assume that the voltage is varied sufficiently slowly such that the number of co-ion layers reaches the equilibrium value, corresponding to the actual value of the voltage.

Referee Comment: 6) The analysis of the MD simulations includes a discussion of adsorption/desorption times (presented in Figure 3b), but these times are never defined.

Reply: We have defined the adsorption/desorption times (page 10 and Supplementary Figs S10 and S11):

To determine the optimal τ , we studied the τ dependence of the co-ion desorption and counter-ion adsorption times, which are the times needed by the co-ion and counter-ion densities to reach their equilibrium values corresponding to the final applied voltage (these times are not the same [42] and depend on charging protocol $U(t)$; see Figs. S10 and S11).

We thank the Referee for spotting this.

Referee Comment: 7) On page 10, it is said that « The data suggest a linear increase of t_{charge} with l in the latter case, consistent with the asymptotic analysis below eq. (6). In the

former case $t_{\text{charge}} \sim l^2$ ». However that the discussion below Eq. (6) applies in specific limits: are they the relevant ones here?

Reply: We have estimated the threshold voltage, which turned out to be about 1.5V for the lengths considered in the simulations, meaning that it does apply in this regime. However, the threshold voltage increases with increasing the pore length and may eventually become larger than the applied voltage (3V). We, therefore, cannot claim any longer that the charging times will increase linearly with the pore length for large lengths. We have clarified the issue with the threshold voltage (page 8):

The latter result is in line with the quadratic pore-length scaling of the charging times of optimal linear sweep [42]. For the pore-lengths considered in the simulations ($l = 12$ nm, 16 nm, and 20 nm), the threshold voltage U_t varies slightly around 1.5 V and increases roughly as a square root of logarithm with increasing l .

We also removed the sentence that the Referee refers to. We are grateful to the Referee for raising this issue!

Referee Comment: 8) On page 11, the authors compare their experimental results with the theoretical predictions of Ref. 51 for the slow time scale, which involves « a salt-concentration dependent factor (of order unity for dilute electrolytes) ». Which value was considered here, since RTILs are not dilute electrolytes?

Reply: As Ref. 51 (now Ref. 56) did not report α for the RTIL simulation in their appendix, for the old version of our manuscript, we used $\alpha=0.3$, which indeed, referred to a dilute electrolyte. For our updated manuscript, we contacted Cheng Lian and he determined $\alpha=0.59$ from Fig. S9 of the Supplementary Material of Ref. 56 for us. Our new calculation uses $\alpha=0.59$ accordingly.

Referee Comment: 9) On page 13, the authors say to motivate this part that « for discharging, there is no danger of clogging the pore entrance either with co-ions or counter-ions. » Could the authors explain why?

Reply: This is because for discharging, the ions experience passive diffusion, while for charging there is active migration (in the bulk of a supercapacitor) driven by the applied potential. We have clarified this in the text (page 13):

For step discharging the ions diffuse passively in and out of the pore, which is in contrast to active migration in the bulk of a supercapacitor as driven by the applied potential during charging. Hence there is no danger of clogging the pore entrance either with co-ions or counter-ions.

Referee Comment: 10) The MD results for the discharge (in particular Figure 5) includes a discussion of the number of co- and counter-ions inside the electrodes. The ions are identical (only with opposite charges) in MD, but this is not the case in real systems, in particular EMIM-BF₄ in the present experiments. Could the authors comment on the effect of

this asymmetry on their optimization procedure? Of course the exact result will be system specific, but it is probably possible to make some general comments (or at least underline the limitations of the present case).

Reply: We expect similar qualitative behaviours. In fact, we have shown in our previous work (Ref. 42) that in the case of charging, co-ion trapping occurs also for BMIM-PF₄, though clearly charging at the cathode and anode proceed asymmetrically. We now emphasize this point in the MS (page 21):

The results of ref. [42] show that co-ion trapping and pore clogging occur for both size-symmetric and size-asymmetric ions and hence linear voltage sweeps can accelerate charging in both cases. Likewise, we expect that non-linear sweeps can speed up charging for asymmetric ions but we leave such studies to further work.

For discharging (Fig. 5 and 6) referred to by the Referee, we again expect a similar behaviour, but the inversion-protocols will be different at the cathode and the anode. In practical applications, however, one can optimize a supercapacitor as a whole. We comment on this issue in the manuscript (page 17):

Although we have used same-size ions in our simulations, we expect similar behaviour for cation and anions with different sizes. In this case, an optimal (U_{inv} , k_{inv}) pair might be different for the cathode and the anode and one may need to compromise the speed of discharging at one of the electrodes. In practice, U_{inv} and k_{inv} must be optimized for the entire supercapacitor.

Referee Comment: 11) *The caption of Figure 6 refers to the optimal (U_{inv} , k_{inv}) pair, but based on the rest of their discussion presumably this optimum depends on the initial charging state, hence U_{ch} . Could the authors discuss this point?*

Reply: Yes the Referee is correct. There is a separate optimal pair for each charged state. We have made it clear in the manuscript (page 17):

This figure clearly demonstrates that there is a global minimum in the discharging times, obtained at $U_{inv} \approx -2.5$ V for our model supercapacitor charged at $U_{ch} = 3$ V (note that an optimal pair (U_{inv} , k_{inv}) depends on U_{ch}).

Referee Comment: 12) *At the bottom of page 15, the authors introduce « the minimum discharging time, defined as the time at which the charge Q vanishes and exhibits only small fluctuations around $Q = 0$ ». How is it determined in practice? Is it only a visual estimate, or is there a more systematic criterion (or algorithm) to measure it from the data?*

Reply: The charge fluctuations were within 5% of the initial charge. We have manually checked this separately for each discharging curve. We have updated the text accordingly and thank the Referee for spotting this (page 16):

the charge Q vanishes and exhibits only small fluctuations around $Q = 0$ within 5% of the initial charge (note that it is possible that the charge crosses zero and then decays to $Q = 0$ from the other side).

Referee Comment: 13) On page 16, it is said that « the values of k_{inv} for the experimental setup had to be taken much larger than k_{inv} used for the single nanopore of the MD simulations. » Isn't it much smaller instead (i.e. much longer timescale)?

Reply: Yes the Referee is correct. We have corrected this typo and thank the Referee for spotting it.

Referee Comment: 14) On page 18, the authors indicate that « work needs to be done to differentiate the linear and non-linear protocols in the regime of full charging. » It would indeed be relevant to discuss (already in the present work) some other considerations than the charging/discharging time, in particular the work and/or power (this last aspect is only mentioned in passing in the Supplementary Material, at the end of section S1C for the RC circuit).

Reply: We feel that it is important that our manuscript stays focussed. We have mentioned several references that already looked at work and power optimization of supercapacitors (Refs. [12, 17–28, 37–43], see page 2). The influence of the time-dependent applied potential on discharging and charging times that we discuss in our paper has received little attention in the supercapacitor literature yet. We now emphasize this in the new version of our manuscript (page 4 [ref3:14]):

In this article, we focus on the second question, which has received much less attention in electrochemistry.

Referee Comment: 15) At the very end of the conclusion, the authors indicate that: « The finding that discharging can be accelerated may find a useful application in capacitive deionization, where for the production of potable water via ion electrosorption, the operation speed is a very important factor. » While this may be true, the electrolyte in this context is very different from the present case (dilute electrolyte solution vs RTILs). It would be interesting to add a comment on how the present findings may hold (or not) for dilute electrolytes, or at the very least necessary to include a caveat on the applicability to these systems.

Reply: Since the voltage-inversion procedure is based on accelerating ion adsorption/desorption via application of a transient potential difference, it is not bound to nanopores and shall be applicable in the case of dilute electrolytes and even flat electrodes. In fact, the idea to apply a voltage-inversion comes from the analysis of an RC circuit, where an inverse potential spike discharges a supercapacitor momentarily (Section S1.C in the Supplementary material). We have added a comment in the manuscript emphasizing this point (page 19):

Although we assessed this procedure with subnanometre pores and neat ionic liquids, it shall work for mesoporous electrodes as well as dilute electrolytes, because it is based on temporary speeding-up of ion adsorption/desorption by applying a transient potential difference (Supplementary Note S1 C).

REVIEWERS' COMMENTS

Reviewer #2 (Remarks to the Author):

The authors correctly answered my comments. I now support the publication of the paper.

Reviewer #3 (Remarks to the Author):

The authors have addressed all my previous comments. I would now be happy to recommend publication, after the authors have clarified a last issue raised by their answer to my previous comment #5. They have added in the manuscript: "We assume that the voltage is varied sufficiently slowly such that the number of co-ion layers reaches the equilibrium value, corresponding to the actual value of the voltage." I think that it is important to elaborate on this key assumption, including an estimate of the equilibration time based on physical parameters. This would allow to compare it to the resulting optimal charging/discharging time to check a posteriori the validity of this assumption. In particular, it would clarify whether this assumption contributes to the differences between molecular simulations and analytical predictions.

Response to the Referee reports

We thank all Referees for the positive assessment of our work. A detailed response to the comments of Referee #3 is provided below. We refer to pages in the revised manuscript, where the changes have been made in response to the Referee's suggestion, and also provide a diff file, wherein all changes are highlighted.

RESPONSE TO REFEREE #3

The authors have addressed all my previous comments. I would now be happy to recommend publication, after the authors have clarified a last issue raised by their answer to my previous comment #5. They have added in the manuscript: "We assume that the voltage is varied sufficiently slowly such that the number of co-ion layers reaches the equilibrium value, corresponding to the actual value of the voltage." I think that it is important to elaborate on this key assumption, including an estimate of the equilibration time based on physical parameters. This would allow to compare it to the resulting optimal charging/discharging time to check a posteriori the validity of this assumption. In particular, it would clarify whether this assumption contributes to the differences between molecular simulations and analytical predictions

We thank the Referee for this final question. In response to it, we have thoroughly revised the paragraphs around eqs. (1) and (2) (see page 5) and we think that the text is now much clearer. As suggested, we also included an estimate of the waiting ('equilibration') times in eq. (1) and of the total time needed to free a nanopore of co-ions based on physical parameters.

To check the validity of the assumptions *a posteriori*, we would have to know the co-ion diffusion coefficient $D_c(u)$, which would allow us to calculate the total charging time using eq. (2). In our simulations, however, we treat D_c as a fitting parameter (via τ , eq. (5)). For sufficiently slow charging (*e.g.*, $\tau \geq 12$ ns for $\ell = 12$ nm), we found that the charging times from the MD simulations are consistent with the charging times given by eq. (4), that is, the pore charging completes at the same time as the voltage reaches its final value (see $\tau = 12$ ns in Fig. 3a and Fig. S11). For lower values of τ , the waiting times Δt_n in eq. (1) are too short, which leads to co-ion trapping and slower charging. We have added a text in

the manuscript explaining this point (page 11). However, we do not consider it as a rigorous *a posteriori* proof of the assumptions made in our phenomenological theory. In our view, with MD simulations that take a particular sweep function as an input we cannot prove that precisely that functional form is optimal.

Nevertheless, we think that the proof of the pudding (our theory) is in the MD data: our proposed nonlinear sweep charges a pore faster than the optimal linear sweep (the current state of the art).